# Application of Graphene and Carbon Nanotubes on Carbon Felt Electrodes for the Electro-Fenton System

**DOI:** 10.3390/ma12101698

**Published:** 2019-05-25

**Authors:** Yi-Ta Wang, Chang-Hung Tu, Yue-Sheng Lin

**Affiliations:** 1Department of Mechanical and Electro-Mechanical Engineering, National I-Lan University, Yilan City 26047, Taiwan; kjh615243henry@gmail.com; 2Department of Mechanical Engineering, National Taiwan University of Science and Technology, Taipei City 10607, Taiwan; b0222086@gmail.com

**Keywords:** electro-Fenton system, cathode, graphene, carbon nanotubes, polyvinylidene difluoride

## Abstract

The electro-Fenton system has the ability to degrade wastewater and has received attention from many researchers. Currently, the core development objective is to effectively increase the degraded wastewater decolorization efficiency in the system. In this study, to improve the electro-Fenton system reaction rate and overall electrical properties, we used polyvinylidene difluoride to fix carbon nanotubes (CNTs) and graphene onto the system cathode (carbon felt electrode), which was then used to process Reactive Black 5 wastewater. Furthermore, we (1) used scanning electron microscopy to observe the structural changes in the electrode surface after modification; (2) used the Tafel curve to determine the electrode corrosion voltage and corrosion rate; and (3) analyzed the azo-dye decolorization level. The results showed that the maximum system decolorization rates of the CNT- and graphene-modified carbon felt electrodes were 55.3% and 70.1%, respectively. These rates were, respectively, 1.2 and 1.5 times higher than that of the unmodified carbon felt electrode, implying that we successfully improved the cathode characteristics. The modified electrode exhibited an improved conductivity and corrosion resistance, which, in turn, improved the system decolorization efficiency. This significantly increased the electro-Fenton system overall efficacy, making it valuable for future applications.

## 1. Introduction

In recent years, industrial dyeing and finishing processes have produced a large amount of wastewater. This wastewater often contains many organic and inorganic pollutants such as pastes, grease, and dye interface activation agents that are difficult for microbes to break down. The electro-Fenton process is one of the advanced oxidation processes (AOPs) for the treatment of wastewater of low to moderate organic strength [1]. When power is supplied, the system reaction commences and electrons (e^−^) and hydrogen ions (H^+^) are generated; they promote the reduction of the dissolved oxygen (O_2_) in the solution into hydrogen peroxide (H_2_O_2_). After ferric ions accept electrons, they are reduced to ferrous ions, which react with the H_2_O_2_ produced by the system. This reaction produces hydroxyl radical (^•^OH) and ferric ions, which are important for degrading wastewater. The reaction of an azo dye with hydroxyl radical can help clean the wastewater [1]. With continuous electrical supply, ferric ions will be reduced once more to ferrous ions, which will continue to react with H_2_O_2_ and produce hydroxyl radicals, forming the electro-Fenton system cycle. The respective chemical reactions (Equations (1)–(3)) are given below [2,3].
2H^+^ + 2e^−^ + O_2_ → H_2_O_2_,(1)
Fe^3+^ + e^−^ → Fe^2+^,(2)
Fe^2+^ + H_2_O_2_ + H^+^ → Fe^3+^ + H_2_O + ^•^OH.(3)

The characteristics of the electrode are the main factors that affect the system reaction efficiency. Selection of the cathode electrode is mainly based on good conductivity, good corrosion resistance, high specific surface area, and high stability [4]. More recently, advances have been made using carbon materials due to their non-toxicity, high specific surface area, good electrical conductivity, and high chemical stability. A variety of conductive carbon materials have been successfully used in the electro-Fenton system [5]. Ganiyu et al. found that CoFe-layered double hydroxide (CoFe-LDH) could be grown on carbon felt, which was then used in the electro-Fenton system. The catalyst could increase the system reaction rate to produce ferrous ions and hydroxyl radicals that would effectively degrade organic pollutants [6]. Carbon nanotube (CNT) structures are formed from carbon atoms in sp^2^-mixed orbitals with carbon-to-carbon double bonds. CNTs are multifunctional and porous, with high conductivity and specific surface area. Moreover, they can assist in electron transfer. Wang et al. indicated that the adsorption capacity of multi-wall carbon nanotubes (MWCNTs) was higher than that of activated carbon (AC) and that the surfaces of CNTs promoted the adsorption of hydrocarbons; therefore, they were suitable for treating contaminants [7]. Graphene has a hexagonal honeycomb lattice arrangement (single-layer 2D crystal film) formed from carbon atoms in the sp^2^ orbital. Graphene can be thought of as having a single layer, 2D, carbon-atom-thick structure formed from carbon atoms and covalent bonds [8]. Graphene has excellent electrochemical characteristics, such as fast electron transfer rates and good conductivity; it exhibits a great potential for use as an electrode material. Tsai et al. used CNT/graphene-modified carbon cloth in microbial fuel cells to effectively increase the power density and reduce the internal resistance [9]. Le et al. reported that their coatings were made of reduced graphene oxide (rGO) on carbon felt. Their results indicated that the charge-transfer resistance for the electrode decreased and the cyclic voltammetry (CV) response increased by ~2.5 times [10].

In recent years, carbon materials with 3D structures have been studied; they can be used in electrochemistry as sensors [11]. To change the morphology of the materials, many researchers have used different processing methods. Furthermore, CNTs and graphene have been used to synthesize composition materials and they have been applied in various fields [12,13,14]. Khoshnevis et al. used floating-catalyst chemical vapor deposition (FC-CVD) to produce CNT fibers and the results showed that for different winding rates, the CNT fibers displayed changes in porosity and density, which further affected the mechanical properties [15]. Ioniță et al. synthesized cellulose acetate (CA) membranes doped with CNTs and graphene oxide (GO) and the membranes exhibited outstanding biological performance [16]. Mousset et al. used graphene with 2D structures and graphene foam with 3D structures as electrodes to treat phenol in the electro-Fenton system. The results showed that for the graphene foam with a porous structure and high specific surface area, the phenol degradation was 2.5 times higher than that of the graphene with a 2D structure [17]. In summary, the combination of CNTs and graphene can change the morphology of the electrode material they are deposited on and influence the properties of the electrode.

According to the literature cited above, CNTs and graphene have been widely used as electrode materials; however, hardly any research has demonstrated the difference between the efficiency of CNTs and graphene material in the electro-Fenton system. In this study, carbon felt was used as the substrate and CNTs and graphene were used to modify the electrodes; this was expected to increase the specific surface area and the property for oxidation–reduction reactions of the cathode in the electro-Fenton system. The effects of the modified carbon felt material on the cathode plates were investigated.

## 2. Materials and Methods

### 2.1. Experimental Procedure

A 250 mL beaker was used as the experiment tank along with an electrochemical workstation with three electrodes. A Pt plate served as the anode. The working electrodes were unmodified, CNT-modified, and graphene-modified carbon felts. Ag/AgCl was the reference electrode. The distance between the working electrode and the counter electrode was 15 mm. The solution in the reaction tank contained 0.1 M potassium nitrate (KNO_3_) and 20 ppm ferrous sulfate (FeSO_4_·7H_2_O). The O_2_ flow rate was 50 sccm. Nitric acid was used to maintain the pH at 3. An air pump was used to introduce oxygen while the solution was magnetically stirred for homogeneity. The optimal potential was −0.65 V for each electrode in this study [17,18]. When measuring the H_2_O_2_ yields for each electrode, the solution only contained 0.1 M KNO_3_ in the tank; titanium(IV) sulfate (Ti(SO_4_)_2_) was used as the reagent that reacted with the H_2_O_2_ produced during the electrogeneration process. The absorbance for the sample mixed with Ti(SO_4_)_2_ and H_2_O_2_ was measured using a visible-light spectrophotometer (wavelength: 410 nm) and the calibration curve was used to determine the concentration of H_2_O_2_ [19]. The Reactive Black 5 (RB 5) azo dye was employed to evaluate the decolorization rate of each electrode using the visible-light spectrophotometer. A wavelength of 595 nm was used to measure the changes in the decolorization level for 30 min [20]. The experimental setup is shown in Figure 1.

### 2.2. Carbon Felt Electrode Modification

A carbon felt electrode (length × width × height: 20 × 40 × 7; units: mm) was used in this study. It was soaked in a H_2_O_2_ and deionized water solution and heated at 90 °C for 3 h to increase the hydrophilicity of the carbon felt. To fabricate a CNT-modified carbon felt electrode, CNT and polyvinylidene difluoride (PVDF) powders were mixed in a 3:20 proportion [21]. On the other hand, to fabricate a graphene-modified carbon felt electrode, 0.1 g of graphene powder was mixed with 2.5 mg of the PVDF powder [22]. After mixing, 50 mL of N-methyl-2-pyrrolidone (NMP) was added and ultrasound agitation was conducted for 1 h to achieve slurries. The slurries were dropped onto the carbon felt using the spin-coating method. Finally, both the electrodes were dried in a vacuum oven at 200 °C. Modified working electrodes were obtained after the NMP solution was evaporated.

## 3. Results and Discussion

### 3.1. Observation of Electrode Surface

Figure 2A shows the scanning electron microscopy (SEM) image of the unmodified carbon felt. The image indicates that the fiber surface of the carbon felt is smooth with nothing adhered to it. Figure 2B shows the SEM image of the CNT-PVDF-mixture–modified carbon felt magnified 7000 times. Attached objects are clearly visible on the fiber surface; they allow the CNTs to successfully adhere to the originally smooth fiber surface. This increases the specific surface area of the carbon felt. Figure 2C shows the SEM image of the G-PVDF-mixture–modified carbon felt magnified 7000 times. Scale-like graphene is clearly visible. This shows that thorough mixing with PVDF causes CNTs and graphene to effectively adhere onto the surface of the carbon felt fiber after 2 h of carbonization at 200 °C. This completes the modification of carbon fibers using high-conductivity graphene in order to increase the specific surface area of the carbon felt. In addition, this improves the conductivity and enhances the system efficiency.

### 3.2. Fourier-Transform Infrared Spectroscopy Analysis

Figure 3 shows the Fourier-transform infrared spectroscopy (FTIR) spectra for the different modifications of the carbon felt. The carbon felt and CNTs were not modified in this study; consequently, there are no significant peaks in Figure 3, which indicates that the carbon felt and CNTs have very few oxygen functional groups. Yuan et al. used tin oxide (SnO_2_) and polypyrrole (PPy) nanotubes to modify CNTs. Before the modification, there were no obvious indicative CNT peaks in the FTIR spectra [23]; this is consistent with the results of this study. Unlike the carbon felt and CNT/C electrodes, the graphene/C electrode presents peaks with weak vibration intensities at 3446, 1620, 1212, and 1065 cm^−1^, which are indicative of O-H, C=C, C-O-C, and C-O bonds, respectively. Graphene only presents a short peak indicative of oxygen functional groups; therefore, we expect it to be hydrophobic for the electrode [24].

### 3.3. Contact Angle Measurement and Analysis

To further analyze the increase in the specific surface area and performance of the modified carbon felt substrate and the carbon felt electrode, we conducted a water-drop contact angle experiment to measure the hydrophilic characteristics. A smaller contact angle signifies better hydrophilic characteristics of the material toward the solution. The water-drop contact angles of the unmodified, graphene-modified, and CNT-modified carbon felt electrodes were determined to be 144.5°, 136.1°, and 78.1°, respectively. The CNT- and graphene-modified electrodes were superior to the untreated carbon felt electrodes. Wang et al. indicated that CNTs exhibited high adsorption capacities [7]. Dhand et al. added CNTs to a PVDF membrane and the contact angle decreased from 103.6° to 88° [25]. On the other hand, Wu et al. pointed out that graphene, as well as the membrane prepared by mixing graphene and PVDF, exhibited hydrophobic properties. The contact angle increased with the concentration of graphene [26]; hence, the contact angle for graphene is less than that for CNTs. Incidentally, Miao et al. used sulfonated graphene oxide (SGO) to modify a PVDF substrate. During the measurements, at 600 s, the contact angle (hydrophilic property) changed from 76.8° to 46.6° due to the SGO, which had oxygen-containing groups on its edge. This effectively changed the thin film’s contact angle, making it more hydrophilic [27].

### 3.4. Analysis of H_2_O_2_ Generation

Linear sweep voltammetry (LSV) results are related to the electric potential change rate and are generally used for studying solid electrodes. The main objective is to determine the scope of the electrochemical reaction. Many reactions involve electron transfer simultaneously accompanied by chemical reactions; consequently, an appropriate electric potential range must be set for scanning, using obtained sufficient information that is beneficial for projecting and analyzing the reaction mechanism. The electric potential scanning rate represents the electrochemical reaction time. Hence, different scanning rates can be used to observe the current and electric potential changes, in order to understand the complex reaction mechanism. For the given cathode, the current responses were driven by oxygen reduction reactions and the 2-electron reaction pathway that generated H_2_O_2_, which are given in Equation (4) and Equation (5), respectively. Therefore, a relatively high net current implies high oxygen reduction reaction activities, which will, in turn, further promote the H_2_O_2_ generation [17]. The optimal potential was in the range of −0.6 to −0.7 V for the carbonaceous electrodes; this has also been reported previously [17,18]. To measure the response current for each electrode, a fixed scanning rate of 1 × 10^−2^ V/s and scanning voltages from 0 to −1 V were used in this experiment. Figure 4A displays the response current for each electrode. These results indicated that the cathode modified with graphene showed the highest response current (−4.31 mA/cm^2^) at −0.65 V due to the enhancement in the conductivity and specific surface area and that it was superior to the CNT-modified (−1.32 mA/cm^2^) and unmodified carbon felt electrodes (−0.56 mA/cm^2^). To confirm that the results of the LSV experiment were accurate, the H_2_O_2_ concentration was measured using the colorimetry method. Figure 4B shows the H_2_O_2_ concentration for different modifications of carbon felt via electrogeneration after 30 min; the graphene/C electrode generated the highest H_2_O_2_ concentration (0.261 mM), which was similar to that generated by the graphene foam electrode with 3D structures prepared by Mousset et al. [17]; the unmodified carbon felt and CNT/C electrodes produced about 0.098 and 0.138 mM of H_2_O_2_ after 30 min, respectively. This implied that the graphene-modified carbon felt plate could stably produce H_2_O_2_ and improve the overall system reaction. In summary, the carbon felt modified with graphene expectedly produced the most H_2_O_2_ and exhibited the best efficiency for RB 5 azo dye treatment.
Anode    2H_2_O → O_2_ + 4H^+^ + 4e^−^(4)
Cathode    2H^+^ + 2e^−^ + O_2_ → H_2_O_2_.(5)

### 3.5. Tafel Curve Analysis

The Tafel curve analysis results indicate that a higher corrosion voltage corresponds to a lower electrode activity, which demonstrates that the electrode has a better corrosion resistance, i.e., it is less easily corroded. A smaller corrosion electric potential indicates a higher activity and probability of corrosion. The corrosion voltage can be obtained at the point where the anode and cathode polarization curves intersect [28]. As shown in Figure 5, the measurement indicates that the corrosion voltage of the graphene-modified carbon felt electrode is 489.582 mV. This is superior to the 432.212 mV of the CNT-modified carbon felt electrode and the 416.857 mV of the unmodified carbon felt electrode. For a lower corrosion rate and higher corrosion resistance of the electrode, the corrosion current could be obtained using the Tafel extrapolation. The results indicated that the corrosion currents of the CNT- and graphene-modified carbon felt electrodes were 1.07 × 10^−7^ and 2.95 × 10^−6^ A, respectively. Owing to the outstanding conductivity of graphene, a path for electrochemical reactions was provided; consequently, the corrosion current of the graphene-modified carbon felt electrode was higher than that of the CNT-modified carbon felt electrode [29]. However, the currents of both the modified electrodes were higher than the 5.75 × 10^−6^ A current of the unmodified carbon felt electrode. Therefore, the graphene-modified carbon felt electrode exhibited excellent anticorrosion properties in the electro-Fenton field and prevented the corrosion of the electrode.

### 3.6. Cyclic Voltammetry Analysis

Cyclic voltammetry is used to determine the electrochemical activities of a system. More significant peak values in the test measurements indicate stronger redox activities of the electrodes. This experiment was performed via the impregnation of a 0.01 M potassium ferricyanide (K_3_[Fe(CN)_6_]) solution and the scan rate was set to 10 mV/s. Figure 6 shows the measurement graph and the order is as follows: Peak current (I_p_; expressed in A) of carbon felt (6.71 mA) < CNT (16.88 mA) < graphene (60.82 mA). Huang et al. coated graphite felt (GF), polypyrrole/ClO_4_^-^, and polypyrrole/lignin on graphite felt and found that the electroactive surface areas were 4.4, 43.1, and 60.8 cm^2^, respectively [30]. Therefore, the experimental results could be used in the Randles–Sevcik equation given in Equation (6) [17,31] to assign the peak current (I_p_) to the electroactive surface area (A) of the electrode, where *n* is the number of electrons participating in the redox reaction (*n* = 1), D is the coefficient of the probe molecule (7.60 × 10^−6^ cm^2^/s), C is the concentration of the probe molecule (1 × 10^−5^ mol/cm^3^), and γ is the scan rate of the potential perturbation (0.01 V/s). The order of the display electrode surface areas was as follows: Carbon felt (9.03 cm^2^) < CNT/C (22.76 cm^2^) < Graphene/C (82.02 cm^2^). The high electroactive surface area could be attributed to the enhancement in the specific surface area; this is consistent with the results of Mousset et al. [17]. Moreover, the high electroactive surface area could be used to predict an increase in the degradation rate in the electro-Fenton system.
I_p_ = 2.69 × 10^5^ × AD ^1/2^ n ^1/2^γ ^1/2^ C.(6)

### 3.7. RB 5 Degradation Level Analysis

In this study, 40 ppm of the RB 5 azo dye was used to evaluate the treatment efficiency in the electro-Fenton system. The experiment was conducted for 30 min. The light absorbance was measured at intervals of 3 min. The unmodified, CNT-modified, and graphene-modified carbon felt electrodes were all analyzed, as shown in Figure 7. The graphene- and CNT-modified carbon felt electrodes were superior to the unmodified carbon felt electrode in terms of RB 5 decolorization in the electro-Fenton system (70.12% and 55.34%, respectively). During the LSV and H_2_O_2_ concentration measurements, a high response current indicated a high H_2_O_2_ production and hydroxyl radical generation, which led to further attack on the -N=N- double bond in the dye. Nam et al. used a Fenton system with ethylenediaminetetraacetic acid (EDTA) to decolorize an azo dye and effectively treat wastewater. The reaction time for decolorizing methyl red was 10 min (decolorization rate of 71%). For Orange I, the reaction time was 2 min (decolorization rate of 98%). This indicated that the Fenton system underwent a superior reaction. They also proposed that excess H_2_O_2_ would not produce good results in the Fenton reaction [32].

The total organic carbon (TOC) was determined using the ultraviolet light–persulfate photochemical oxidation method to analyze the organic matter content in the sewage. The TOC calculation method was based on Equation (7), where ΔTOC is the change in the concentration response and TOC_0_ (mg/L) is the initial value of total organic carbon. For the graphite felt (GF), the taffeta carbon fiber (TCF) removal of pyrimethanil indicated a TOC removal at 0.3 A of up to 42.36% after 120 min of the electro-Fenton reaction [33]. Zhou et al. used CF to decompose *p*-nitrophenol contaminants in the electro-Fenton system, with a TOC removal rate of 22.2% in 30 min and a CF treated with ethanol and hydrazine hydrate (N_2_H_4_·H_2_O) TOC removal rate of 51.4% [34]. This implied that the treated carbon felt in the electro-Fenton system could effectively reduce the organic pollutants in the sewage. Therefore, an experiment on the degradation of organic pollutants in the electro-Fenton process was performed using graphene-modified, CNT-modified, and unmodified carbon felt electrodes. The respective TOC removal rates were 55.56%, 50.13%, and 10.60% after 30 min. Due to the complexity of the RB 5 molecule, the TOC removal rate was lower than the decolorization rate in the electro-Fenton system [35], as shown in Table 1. The results obtained for the modified carbon felts were more than five times higher than those for the unmodified carbon felt. The modified electrodes could effectively increase the system reaction rate and degrade the organic pollutants, thereby purifying the wastewater. According to the LSV and CV analyses, the high response current and electroactive surface area improved the degradation rate in the electro-Fenton system. These results indicated that the graphene-modified electrode could produce high quantities of H_2_O_2_, which is consistent with the TOC analysis results. The system produced large quantities of hydroxyl radicals and broke down the azo dye, thereby achieving wastewater purification.
TOC removal (%) = (ΔTOC/TOC_0_) × 100.(7)

## 4. Conclusions

In this study, carbon felt electrodes were modified with CNTs and graphene and their characteristics as electro-Fenton system cathodes were investigated by processing RB 5 wastewater. 

The following conclusions were drawn.
The surface morphology showed that the CNT and graphene modifications of the carbon felt electrodes led to the roughening of the smooth carbon felt fiber surface. This assisted in improving the specific surface area and conductivity of the carbon felt. The contact angles indicated that the CNT- and graphene-modified carbon felt electrodes exhibited higher hydrophilicity. This effectively improved the tank electrode and solution reaction time in the electro-Fenton system.The graphene-modified carbon felt exhibited the best reaction rate and electrochemical activity, based on the LSV and CV tests. A high current indicated the enhancement in the response of the electro-Fenton system. The Tafel curve showed that the corrosion resistance of the electrode was also improved by the modification; the modified carbon felt electrodes exhibited superior corrosion resistance compared to the conventional carbon felt plate.When a working electrode was applied to degrade the RB 5 azo dye in the electro-Fenton system, the decolorization rate of the CNT-modified carbon felt electrode was 55.3% and that of the graphene-modified carbon felt electrode was 70.1%, which were, respectively, 1.2 and 1.5 times higher than that of the unmodified carbon felt electrode. A TOC-removal experiment proved that the modified electrode could effectively degrade organic pollutants and improve the system efficiency.

The results indicate that graphene-modified electrodes can produce a high quantity of H_2_O_2_, which is consistent with the TOC analysis results. The system produced large quantities of hydroxyl radicals and broke down the azo dye, thereby achieving wastewater purification. Assuming that the additive quantities were equal, the graphene-modified carbon felt was superior to CNT-modified carbon felt. Both the modified felts exhibited effectively increased conductivity and corrosion resistance and both improved the degrading efficiency of the cathode.

## Figures and Tables

**Figure 1 materials-12-01698-f001:**
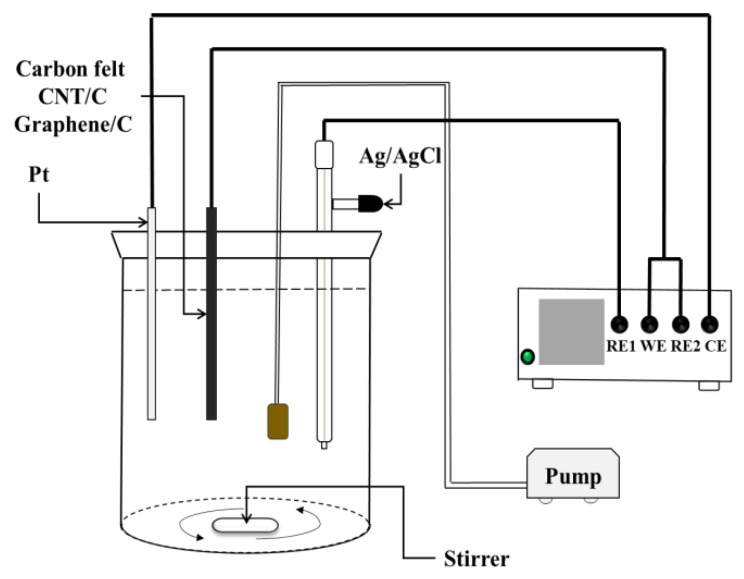
Schematic illustration of the electro-Fenton system.

**Figure 2 materials-12-01698-f002:**
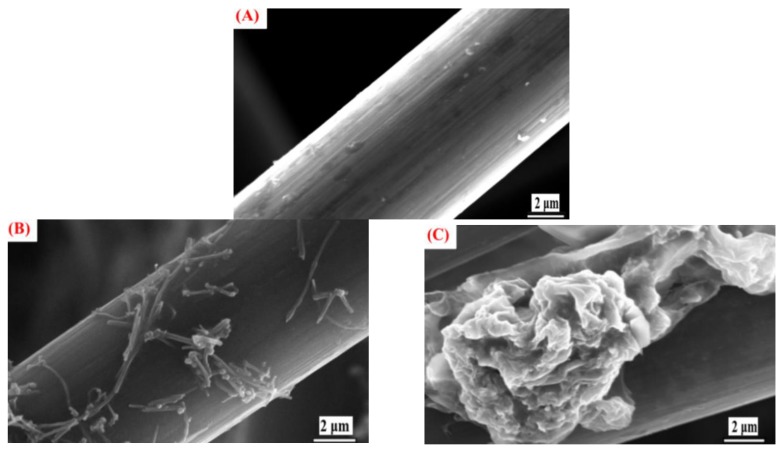
SEM images of carbon felt: (**A**) Unmodified carbon felt magnified 7000 times, (**B**) carbon nanotube (CNT)/C magnified 7000 times, and (**C**) graphene/C magnified 7000 times.

**Figure 3 materials-12-01698-f003:**
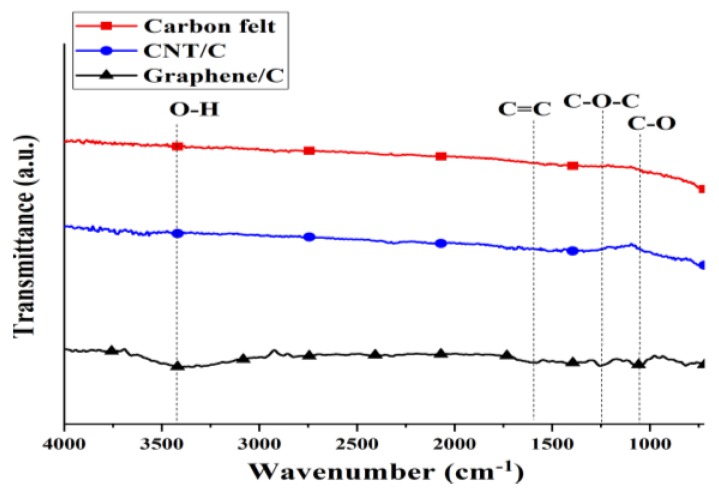
FTIR spectra for different modifications of carbon felt.

**Figure 4 materials-12-01698-f004:**
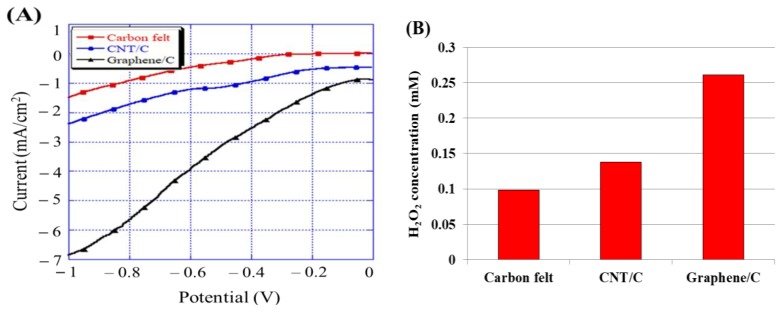
(**A**) Linear sweep voltammetry curves for different modifications of carbon felt; (**B**) H_2_O_2_ concentration for different modifications of carbon felt via electrogeneration after 30 min (−0.65 V).

**Figure 5 materials-12-01698-f005:**
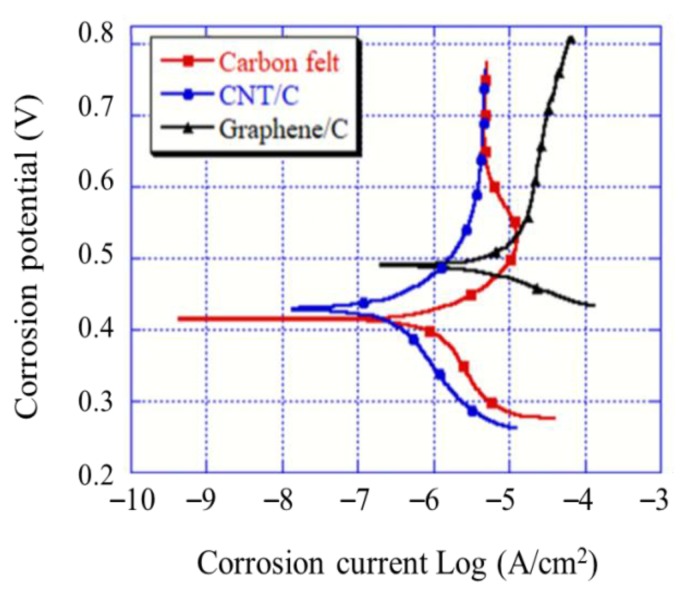
Tafel curves for different modifications of carbon felt.

**Figure 6 materials-12-01698-f006:**
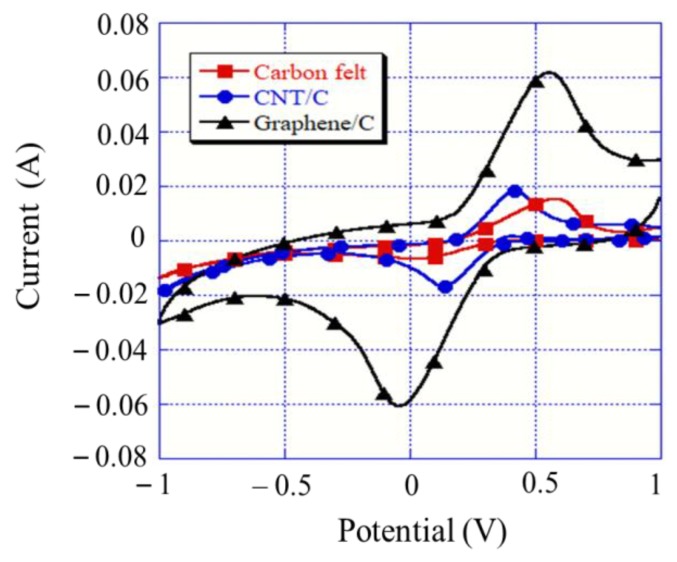
Cyclic voltammetry curves for different modifications of graphite felt.

**Figure 7 materials-12-01698-f007:**
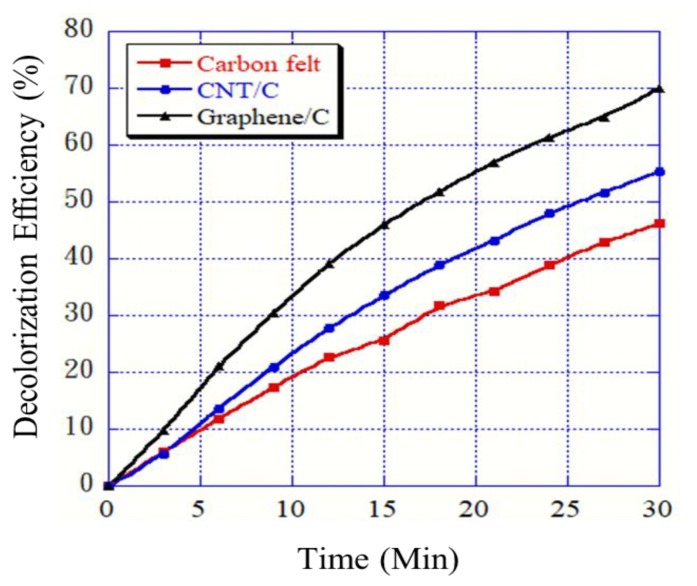
Plots of decolorization rates for different modifications of carbon felt in electro-Fenton system.

**Table 1 materials-12-01698-t001:** Decolorization efficiency and total organic carbon (TOC) removal after 30 min of the electro-Fenton treatment proposed in this work (40 ppm Reactive Black 5 (RB 5)).

Cathode	Carbon Felt	CNT/C	Graphene/C
Response Current (−0.65 V)	−0.56 mA/cm^2^	−1.32 mA/cm^2^	−4.31 mA/cm^2^
H_2_O_2_ yield	0.098 mM	0.138 mM	0.261 mM
Electroactive surface area	9.03 cm^2^	22.76 cm^2^	82.02 cm^2^
Decolorization efficiency	46.15%	55.34%	70.12%
TOC removal	10.60%	50.13%	55.56%

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
