# Peer review of "Application of Graphene and Carbon Nanotubes on Carbon Felt Electrodes for the Electro-Fenton System"

_materials, 2019, doi:10.3390/ma12101698_

Round 1

Reviewer 1 Report

In this research work authors investigated the synthesis, the characterization and treatment performance of two composite carbon-based materials as cathodic electrodes in electro-Fenton systems, for dying wastewater decolorization. Although this is a well-thought study, the language, the organization of the material, the presentation and justification of results are very poor. The reviewer was confused and rather tired by the "wooden" language, the repetitions, the lack of a systematic experimental design, the irrelevant comments/justifications and the many assumptions. Authors are advised to drastically correct the English language so as to meet the standards for publication. In addition, a systematic experimental design should be carried out, investigating important operating variables (current density, pH, etc.) on process performance (as shown in Table 1 single exp. conditions are applied). Other remarks, not necessarily in order of importance are as follows:

- Introduction: This section should be drastically enhanced. The attributes of electro-Fenton over the conventional Fenton process should be clearly described, as well as, the respective developments in the cathodic electrode materials, with emphasis on the 3-D carbon-based ones. Important review and research papers in this field are missing. 

- All Tables are short and not important to the discussion. Alternatively, authors can merge experimental conditions and results in one Table.

- Lines 149-153: Remove repetition.

-Fig. 2: The SEM image of the unmodified carbon felt is missing.

- The LSV scope of analysis is blur and not conclusive for the scope of this study.

-Lines 196-197: How is this conclusion justified? No H2O2 measurements are provided here (this is a serious short-coming).

- Line 197: Correct "Fig. 3".

Author Response

Cover letter to editor

JournalMaterials (ISSN 1996-1944)

Manuscript IDmaterials-488606

Manuscript title: Application of Graphene Carbon Felt Electrodes in the Electro-Fenton System

TypeArticle

Submitted to SectionAdvanced Composites

Dear Editor

We appreciate the reviewers’ valuable comments and the editor’s help. The manuscript has been revised seriously according to the comments item by item in the following. To clearly show the revisions, the revised manuscript keeping the “Track Changes” functions of MS word.

The original paper title “Application of Graphene Carbon Felt Electrode In Electro-Fenton System” has been slightly revised as “Application of Graphene Carbon Felt Electrodes in the Electro-Fenton System”.

The manuscript has been extensively edited in the language, punctuation, grammar, expression clarity, and flow by the professional English editing company “editage Co.”. An English editing certificate for editage Co. is attached to the end of this cover letter.

We surely believe that the quality of the paper will satisfy the requirements of this “Materials”. Your kindest consideration will be greatly appreciated.

I am looking forward to hearing from you.

With best wishes,

Yours sincerely

Yi-Ta Wang

Yi-Ta Wang

Division Director, Center for Nano Science & Technology.

Associate Professor,

Department of Mechanical and Electromechanical Engineering, National Ilan University NIU.

No. 1, Sec. 1, Shennong Rd., Yilan City, Yilan County 26047, Taiwan, R.O.C.

Tel: +886-3-9317457; +886-3-9317082

Reply to the Reviewer's comments:

Reviewer 1

Comment 1: In this research work authors investigated the synthesis, the characterization and treatment performance of two composite carbon-based materials as cathodic electrodes in electro-Fenton systems, for dying wastewater decolorization. Although this is a well-thought study, the language, the organization of the material, the presentation and justification of results are very poor. The reviewer was confused and rather tired by the "wooden" language, the repetitions, the lack of a systematic experimental design, the irrelevant comments/justifications and the many assumptions. Authors are advised to drastically correct the English language to meet the standards for publication. In addition, a systematic experimental design should be carried out, investigating important operating variables (current density, pH, etc.) on process performance (as shown in Table 1 single exp. conditions are applied). Other remarks, not necessarily in order of importance are as follows:

 Reply 1We appreciate the reviewer’s comments. The manuscript has been extensively edited in the language, punctuation, grammar, expression clarity, and flow by the professional English editing company “editage Co.”, and all the contents in the manuscript has been revised seriously according to the comments item by item. We surely believe that the quality of the paper will satisfy the requirements of this journal. 

The optimal potential was -0.65 V for the each electrode in this study [17, 18]; the lower potential will not provide enough power to react. Conversely, higher potential will cause the presence of competitive parasitic reactions, including H2O2 decomposition and H2 evolution. 

Most of the studies reported that the optimum pH of Fenton process is around 3. Acidic medium is the favorable condition for the production of H2O2. Due to the regeneration of Fe2+, through reaction between Fe3+ and H2O2, Fenton process becomes less effective at pH < 3. At higher pH, the efficiency of E-Fenton process decreases rapidly, especially pH > 5. This is due to the fact that H2O2 is unstable in basic solution. H2O2 rapidly decomposes to oxygen and water at neutral to high pH. (Nidheesh, P.V.; Gandhimathi, R. Trends in electro-Fenton process for water and wastewater treatment: An overview. Desalination. 2012, 299, 1-15). The Table 1 has been revised adding the citations [17, 18] in the revision manuscript.

 Ref

[17] Mousset, E.; Wang, Z.; Hammaker, J.; Lefebvre, O. Physico-chemical properties of pristine graphene and its performance as electrode material for Electro-Fenton treatment of wastewater. Electrochim. Acta. 2016, 214, 217-230.[18] Wang, Y.; Liu, Y.; Wang, K.; Song, S.; Tsiakaras, P.; Liu, H. Preparation and characterization of a novel KOH activated graphite felt cathode for the Electro-Fenton process. Appl. Catal. B 2015, 165, 360-368

Comment 2: Introduction: This section should be drastically enhanced. The attributes of electro-Fenton over the conventional Fenton process should be clearly described, as well as, the respective developments in the cathodic electrode materials, with emphasis on the 3-D carbon-based ones. Important review and research papers in this field are missing.

Reply 2Thanks for the reviewer’s comment. The introduction section has been rewritten and modified, the discussions for “Fenton process” and “cathodic electrode materials” have been clearly described, and the 3-D carbon-based has been revised in the revision manuscript (line 69-94).

Comment 3: All Tables are short and not important to the discussion. Alternatively, authors can merge experimental conditions and results in one Table.

Reply 3Thanks for the reviewer’s comment. According to the comment, it has been discussion, and the electrode data for LSV, CV, decolorization and TOC removal have been merged in the revision manuscript Table. 5 (line 315-316).

Comment 4: Lines 149-153: Remove repetition.

Reply 4Thanks for the reviewer’s comment. It has been revised in the revision manuscript (line 152-156).

Comment 5: Fig. 2: The SEM image of the unmodified carbon felt is missing.

Reply 5Thanks for the reviewer’s comment. The SEM image of the unmodified carbon felt has been showed in line 172.

 Comment 6: The LSV scope of analysis is blur and not conclusive for the scope of this study.

Reply 6Thanks for the reviewer’s comment. The optimal potential was in the range of -0.6 V to -0.7 V for the carbonaceous electrodes, which had been reported in previous studies [17, 18]; to ensure the data and avoid the unstable response for each electrode, the scanning voltage from 0 to -1 V were used in this experiment. the content for this section has been revised (line 226-230) in the revision manuscript “

This result indicated that the cathode electrode modified with graphene had the highest response current (-4.31 mA/cm2) at -0.65 V due to the enhancement of the conductivity and specific surface area, and that it was superior to the CNT-modified carbon felt electrode (-1.32 mA/cm2) and the carbon felt electrode (-0.56 mA/cm2). This implied that the graphene-modified carbon felt plate can stably produce hydrogen peroxide and improve the overall system reaction [17, 18, 27]”. It has been revised in the revision manuscript(line 226-230).

Ref

[17] Mousset, E.; Wang, Z.; Hammaker, J.; Lefebvre, O. Physico-chemical properties of pristine graphene and its performance as electrode material for Electro-Fenton treatment of wastewater. Electrochim. Acta. 2016, 214, 217-230.

[18] Wang, Y.; Liu, Y.; Wang, K.; Song, S.; Tsiakaras, P.; Liu, H. Preparation and characterization of a novel KOH activated graphite felt cathode for the Electro-Fenton process. Appl. Catal. B 2015, 165, 360-368

Comment 7: Lines 196-197: How is this conclusion justified? No H2O2 measurements are provided here (this is a serious short-coming).

Reply 7Thanks for the reviewer’s comment. The electro-Fenton process is based on the continuous electro-generation of H2O2 at a suitable cathode fed with O2 or air, along with the addition of an iron catalyst to the treated solution to produce •OH. The continuous generation of H2O2 requires the electrons which need to be supplied by an external electricity source. From Equation (1~2), it is known that H2O2 and ferrous ions are generated by electrode reactions, and both directly generate Fenton reactions (such as Equation 3). The reaction rate is very fast. We tried to analyze the H2O2 and ferrous ions in the reaction solution in the experiment. However, they are limited in concentration so that they cannot be accurately measured to their effective amount. Therefore, the present work uses the decolorization rate of the dye compound to determine the efficiency of the electro-Fenton system.

For the given cathode, the current responses were driven by oxygen reduction reactions and the 2-electron reaction pathway that generates hydrogen peroxide, as given in Eq. (4-5). Therefore, a relatively high net current implies high oxygen reduction reaction activities, which will, in turn, promote the hydrogen peroxide generation in the Electro-Fenton system; It has been revised in the revision manuscript (line 218-223).

Comment 8: Lines 197, Correct "Fig. 3".

Reply 8Thanks for the reviewer’s comment. It has been revised in the revision manuscript (line 225).

Reviewer 2 Report

The introduction section must be improved with recent data related to the application of carbon nanotubes, respectively graphene, like Materials Today Energy, 9, 154-186, 2018; Carbohydrate Polymers, 183, 50-61, 2018; Nanomaterials, 9(3), 439, 2019; Materials 12(5), 729, 2019. I also strongly recommend at least one structural analysis method for the modified electrode (FT-IR, Raman, XPS) and the discussion of interactions or the citation of previous reported synthesis by same authors.

Author Response

Cover letter to editor

JournalMaterials (ISSN 1996-1944)

Manuscript IDmaterials-488606

Manuscript title: Application of Graphene Carbon Felt Electrodes in the Electro-Fenton System

TypeArticle

Submitted to SectionAdvanced Composites

Dear Editor

We appreciate the reviewers’ valuable comments and the editor’s help. The manuscript has been revised seriously according to the comments item by item in the following. To clearly show the revisions, the revised manuscript keeping the “Track Changes” functions of MS word.

The original paper title “Application of Graphene Carbon Felt Electrode In Electro-Fenton System” has been slightly revised as “Application of Graphene Carbon Felt Electrodes in the Electro-Fenton System”.

The manuscript has been extensively edited in the language, punctuation, grammar, expression clarity, and flow by the professional English editing company “editage Co.”. An English editing certificate for editage Co. is attached to the end of this cover letter.

We surely believe that the quality of the paper will satisfy the requirements of this “Materials”. Your kindest consideration will be greatly appreciated.

I am looking forward to hearing from you.

With best wishes,

Yours sincerely

Yi-Ta Wang

Yi-Ta Wang

Division Director, Center for Nano Science & Technology.

Associate Professor,

Department of Mechanical and Electromechanical Engineering, National Ilan University NIU.

No. 1, Sec. 1, Shennong Rd., Yilan City, Yilan County 26047, Taiwan, R.O.C.

Tel: +886-3-9317457; +886-3-9317082

Reply to the Reviewer's comments:

Reviewer 2

Comment 1:The introduction section must be improved with recent data related to the application of carbon nanotubes, respectively graphene, like Materials Today Energy, 9, 154-186, 2018; Carbohydrate Polymers, 183, 50-61, 2018; Nanomaterials, 9(3), 439, 2019; Materials 12(5), 729, 2019.

Reply 1Thanks for the reviewer’s comment. The references have been added to the introduction section in the revision manuscript, and we show the position for each reference item by item in the following:

1.   The reference Muhulet, A.; Miculescu, F.; Voicu, S.I.; Schütt, F.; Thakur, V.K.; Mishra, Y.K. Fundamentals and scopes of doped carbon nanotubes towards energy and biosensing applications. Mater. Today Energy 2018, 9, 154-186has been revised in revision manuscript (line 82-84), and the number of citations is [14].

2.   The reference Ioniță, M.; Crică,L.E.; Voicu, S.I.; Dinescu, S.; Miculescu, F.; Costache, M.; Iovu, H. Synergistic effect of carbon nanotubes and graphene for high performance cellulose acetate membranes in biomedical applications. Carbohydr. Polym. 2018, 183, 50-61has been revised in revision manuscript (line 87-89), and the number of citation is [16].

3.   The reference Wang, Y.; Pan, C.; Chu, W.; Vipin, A.K.; Sun, L. Environmental remediation applications of carbon nanotubes and graphene oxide: Adsorption and catalysis. Nanomaterials 2019, 9, 439has been revised in revision manuscript (line 69-72, 200-201), and the number of citation is [7].

4.   The reference Liu, Y.; Wang, Y. Size-dependent free vibration and buckling of three-dimensional graphene foam microshells based on modified couple stress theory. Materials 2019, 12, p. 729has been revised in revision manuscript (line 81-82), and the number of citation is [11].

Comment 2I also strongly recommend at least one structural analysis method for the modified electrode (FT-IR, Raman, XPS) and the discussion of interactions or the citation of previous reported synthesis by same authors.

Reply 2Thanks for the reviewer’s comment. We have used “FTIR” to analyze the electrode with different modification, and the result has been revised in the revision manuscript (line 176-190, Section 3.2).

Reviewer 3 Report

The authors synthesized carbon nanotube (CNT) and graphene carbon felt electrodes for processing Reactive Black 5 (RB5) wastewater. The results show that the graphene carbon felt had the best performance in reaction rate of electrochemical activity, corrosion resistances and good response of the electro-Fenton system. As a result, their decolorization rate for RB5 could reach 70.1%. The work is interesting and can be published in Materials if the following issues can be addressed:

1- The authors should cite the papers “Effect of alignment and packing density on the stress relaxation process of carbon nanotube fibers spun from floating catalyst chemical vapor deposition method” conducted by Hamed Khoshnevis et al. (Colloids and Surfaces A: Physicochemical and Engineering Aspects, 2018, 558, 570-578) and  “Direct Spinning of Horizontally Aligned Carbon Nanotube Fibers and Films From the Floating Catalyst Method” conducted by Hai M. Duong et al. (Nanotube Superfiber Materials (Second Edition): William Andrew Publishing; 2019. p. 3-29.) in the introduction section for better review of application of carbon nanomaterials.

2- Introduction needs to be shortened and improved.

3- The citation style is not properly presented in the manuscript. The authors should correct them.

4- Section 2.2 needs to be improved (ex. Combining the 2 paragraphs of the manufacturing steps of CNT and graphene-modified carbon felt electrode….)

5- Section 3.2 was not analyzed properly. The authors should improve it.

6- There are no clear discussion on how the performance of graphene- modified carbon felt have better performance than the others in whole section 3: Why did CNT/C have the lowest contact angle? why does graphene/C have better results in Linear Sweep Voltammetry Analysis, Tafel curve analysis, Cyclic voltammetry Analysis, Decolorization Level Analysis, and Total organic carbon Analysis. The authors should clarify them and improve the manuscript.

7-Several writing error and grammar need to be corrected. English needs to be polished

Author Response

Cover letter to editor

JournalMaterials (ISSN 1996-1944)

Manuscript IDmaterials-488606

Manuscript title: Application of Graphene Carbon Felt Electrodes in the Electro-Fenton System

TypeArticle

Submitted to SectionAdvanced Composites

Dear Editor

We appreciate the reviewers’ valuable comments and the editor’s help. The manuscript has been revised seriously according to the comments item by item in the following. To clearly show the revisions, the revised manuscript keeping the “Track Changes” functions of MS word.

The original paper title “Application of Graphene Carbon Felt Electrode In Electro-Fenton System” has been slightly revised as “Application of Graphene Carbon Felt Electrodes in the Electro-Fenton System”.

The manuscript has been extensively edited in the language, punctuation, grammar, expression clarity, and flow by the professional English editing company “editage Co.”. An English editing certificate for editage Co. is attached to the end of this cover letter.

We surely believe that the quality of the paper will satisfy the requirements of this “Materials”. Your kindest consideration will be greatly appreciated.

I am looking forward to hearing from you.

With best wishes,

Yours sincerely

Yi-Ta Wang

Yi-Ta Wang

Division Director, Center for Nano Science & Technology.

Associate Professor,

Department of Mechanical and Electromechanical Engineering, National Ilan University NIU.

No. 1, Sec. 1, Shennong Rd., Yilan City, Yilan County 26047, Taiwan, R.O.C.

Tel: +886-3-9317457; +886-3-9317082

Reply to the Reviewer's comments:

Reviewer 3

Comment 1: The authors synthesized carbon nanotube (CNT) and graphene carbon felt electrodes for processing Reactive Black 5 (RB5) wastewater. The results show that the graphene carbon felt had the best performance in reaction rate of electrochemical activity, corrosion resistances and good response of the electro-Fenton system. As a result, their decolorization rate for RB5 could reach 70.1%. The work is interesting and can be published in Materials if the following issues can be addressed:

Reply 1Thanks for the reviewer’s positive comment; we reply the review comments item by item in the following.

Comment 2: The authors should cite the papers “Effect of alignment and packing density on the stress relaxation process of carbon nanotube fibers spun from floating catalyst chemical vapor deposition method” conducted by Hamed Khoshnevis et al. (Colloids and Surfaces A: Physicochemical and Engineering Aspects, 2018, 558, 570-578) and  “Direct Spinning of Horizontally Aligned Carbon Nanotube Fibers and Films From the Floating Catalyst Method” conducted by Hai M. Duong et al. (Nanotube Super fiber Materials (Second Edition): William Andrew Publishing; 2019. p. 3-29.) in the introduction section for better review of application of carbon nanomaterials.

Reply 2The references have been added to the introduction section for this manuscript, and we show the position for each reference item by item in the following:

1.   The reference Khoshnevis, H.; Tran, T.Q.; Mint, S.M.; Zadhoush, A.; Duong, H.M.; Youssefi, M. Effect of alignment and packing density on the stress relaxation process of carbon nanotube fibers spun from floating catalyst chemical vapor deposition method. Colloids Surf. A Physicochem. Eng. Asp. 2018, 558, 71-74,has been revised in revision manuscript (line 84-87), and the number of citation is [15].

2.   The reference Duong, H.M.; Tran, T.Q.; Kopp, R.; Myint, S.M.; Peng, L. Direct spinning of horizontally aligned carbon nanotube fibers and films from the floating catalyst method. Nanotube Super fiber Materials (Second Edition, William Andrew Publishing, 2019, 3-29has been added in revision manuscript (line 83-84), and the number of citation is [12].

Comment 3: Introduction needs to be shortened and improved.

Reply 3 Thanks for the reviewer’s comment; the content for introduction section has been shortened and improved, as below: “The characteristics of the electrode are the main factors that affect the system reaction efficiency. Selection of the cathode electrode is mainly based on good conductivity, good corrosion resistance, high specific surface area, and high stability [4]. More recently, advances have been made using carbon materials due to their non-toxicity, high specific surface area, good electrical conductivity, and high chemical stability. A variety of conductive carbon materials have been successfully used in the electro-Fenton system [5]. Ganiyu found that CoFe-LDH could be grown on carbon felt, which was used in the electro-Fenton system. The catalyst could increase the system reaction rate to produce ferrous ions and hydroxyl radicals that would effectively degrade organic pollutants [6]. Carbon nanotube (CNT) structures are formed with carbon atoms in sp2 mixed orbitals with a carbon-to-carbon double bond. CNTs are multifunctional and porous, with high conductivity, and high specific surface area. Moreover, they can assist in electrons transfer. Wang indicated that the adsorption capacities of multi-wall carbon nanotubes (MWCNTs) are higher than that of activated carbon (AC), and the surface of CNT promotes the absorption of hydrocarbons; therefore, it is suitable for treating contaminants [7]. Graphene has a hexagonal honeycomb lattice arrangement (single-layer 2D crystal film) formed from carbon atoms in the sp2 orbital. Graphene can be thought of as having a single layer 2D carbon-atom-thick structure formed from carbon atoms and covalent bonds [8]. Graphene has excellent electrochemical characteristics, such as fast electron transfer rates and good conductivity; it has a high potential to be used as an electrode material. Tsai et al. used CNT/graphene modified carbon cloth in microbial fuel cells to effectively increase the power density and reduce internal resistances [9]. Le reported that the coatings were made of reduced graphene oxide (rGO) on carbon felt. The results indicated that the charge-transfer resistance for the electrode was decreased, and the CV response was increased by ~ 2.5 times [10]. In recent years, carbon materials with 3D structures have been studied; they can be used in electrochemistry as sensors [11]. To change the morphology of the materials, many researchers have used different processing methods. Besides, CNTs and graphene were used to synthesize the composition material, and they have been applied in various fields [12-14]. Khoshnevis used floating catalyst chemical vapor deposition (FC-CVD) to produce CNT fibers, and the results showed that with different winding rates, the CNT fiber displayed changes in porosity and density, which affected the mechanical property further [15]. Ioniță synthesized cellulose acetate (CA) membranes doped with CNTs and graphene oxide (GO), and the membranes exhibited outstanding biological performances [16]. Mousset used graphene with 2D structures and graphene foam with 3D structures as electrodes to treat phenol in the electro-Fenton system. The results showed that for the graphene foam with a porous structure and high specific surface area, the phenol degradation was 2.5 times higher than that of the graphene with a 2D structure [17]. In summary, the electrodes prepared by compositing the CNTs and graphene can change the morphology of the material, and influence the property of electrodes.”. It has been revised in the revision manuscript (line 33-113).

Comment 4: The citation style is not properly presented in the manuscript. The authors should correct them.

Reply 4 Thanks for the important comment, all the citation style have been checked and corrected carefully in the revision manuscript (line 352-445).

Comment 5: Section 2.2 needs to be improved (ex. Combining the 2 paragraphs of the manufacturing steps of CNT and graphene-modified carbon felt electrode….)

Reply 5Thanks for the reviewer’s comment. The Section 2.2 have been revised “The carbon felt electrode (length x width x height: 20 × 40 × 7; unit: mm) was used as the electrode in this study. The carbon felt electrode was soaked in a hydrogen peroxide and deionized water solution and heated at 90 for 3 h to increase the hydrophilicity of the carbon felt. For the CNT-modified carbon felt electrode formation, the CNT and polyvinylidene difluoride (PVDF) powders were mixed in a 3:20 proportion [20]. Conversely, 0.1 g of graphene powder was mixed with 2.5 mg of the PVDF powder [21]. After mixing, 50 mL of N-methyl-2-pyrrolidone (NMP) was added, after which ultrasound agitation was conducted for 1 h to afford slurries. The slurries were dropped on the carbon felt by the spin coating method. Finally, both of the electrodes were dried in a vacuum oven at 200 . Modified working electrodes were obtained after the NMP solution was evaporated.It has been revised in the revision manuscript (line 135-144).

Comment 6: Section 3.2 was not analyzed properly. The authors should improve it.

Reply 6Thanks for the reviewer’s comment. The content for this section in this manuscript has been improved “Wang indicated in his research that the CNTs had high adsorption capacities [7]. Dhand added CNT to the PVDF membrane and the contact angle reduced from 103.6° to 88° [24]. Conversely, Wu pointed out that graphene, as well as the membrane prepared by mixing graphene and PVDF exhibited hydrophobic properties. The contact angle increases as the concentration of graphene increases [25]; thus, its contact angle is less than that of CNT. Incidentally, Miao used sulfonated graphene oxide (SGO) to change the PVDF. During the measurements, at 600 s, the hydrophilic property changed from 76.8° to 46.6°, due to the SGO, which had oxygen-containing groups in its edge. This effectively changed the thin film’s contact angle, making it more hydrophilic [26].” It has been revised in the revision manuscript (line 200-208).

Comment 7: There are no clear discussions on how the performance of graphene- modified carbon felt have better performance than the others in whole section 3: Why did CNT/C have the lowest contact angle? why does graphene/C have better results in Linear Sweep Voltammetry Analysis, Tafel curve analysis, Cyclic voltammetry Analysis, Decolorization Level Analysis, and Total organic carbon Analysis. The authors should clarify them and improve the manuscript.

Reply 7 Thanks for the important and useful comment, we reply the review comments item by item in the following.

1.     Contact angle: the content for this section has been revised (line 200-204) in the revision manuscript “Wang indicated in his research that the CNTs had high adsorption capacities [7]. Dhand added CNT to the PVDF membrane and the contact angle reduced from 103.6° to 88° [24]. Conversely, Wu pointed out that graphene, as well as the membrane prepared by mixing graphene and PVDF exhibited hydrophobic properties. The contact angle increases as the concentration of graphene increases [25]”

2.     Linear Sweep Voltammetry Analysis: the content for this section has been revised (line 226-230) in the revision manuscript “This result indicated that the cathode electrode modified with graphene had the highest response current (-4.31 mA/cm2) at -0.65 V due to the enhancement of the conductivity and specific surface area, and that it was superior to the CNT-modified carbon felt electrode (-1.32 mA/cm2) and the carbon felt electrode (-0.56 mA/cm2). This implied that the graphene-modified carbon felt plate can stably produce hydrogen peroxide and improve the overall system reaction [17, 18, 27].”

3.     Tafel curve analysis: the content for this section has been revised (line 246-252) in the revision manuscript “Due to the outstanding conductivity for graphene, a path for electrochemical reactions is provided; consequently, the corrosion current of the graphene-modified carbon felt electrode was higher than that of the CNT-modified carbon felt electrode [29]. However, the currents of both of the modified electrodes were superior to the 5.75×10-6 A of the unmodified carbon felt electrode. Thus, the graphene-modified carbon felt electrode could achieve excellent anti-corrosion in the electro-Fenton field and prevent the corrosion of the electrode.” 

4.     Cyclic voltammetry Analysis: the content for this section has been revised (line 268-272) in the revision manuscript “The order of the display electrode was as follows: C (9.03 cm2) < CNT/C (22.76 cm2) < G/C (82.02 cm2). The high electroactive surface area could be attributed to the enhancement of the specific surface area, which was consistent with the research results by Mousset [17]. Moreover, the high electroactive surface area can predict an increase in the degradation rate in the electro-Fenton system.”

5.     RB 5 Degradation Level Analysis: the content for this section has been revised (line 282-284, 309-314) “In the LSV test, the high response current indicated a high hydrogen peroxide production and generated more hydroxyl free radicals to further attack the -N=N- double bond in the dye.” and “For LSV and CV analyses, the high response current and electroactive surface area improved the degradation rate in the electro-Fenton system. The results indicated that the graphene-modified electrode could produce high quantities of hydrogen peroxide, which was consistent with the total organic carbon analysis. The system produced large quantities of hydroxyl free radicals and broke down azo dye, thereby achieving the wastewater purification objective” 

Comment 8: Several writing error and grammar need to be corrected. English needs to be polished.

Reply 8We appreciate the reviewer’s comments. The manuscript has been extensively edited in the language, punctuation, grammar, expression clarity, and flow by the professional English editing company “editage Co.”, and all the contents in the manuscript has been revised seriously according to the comments item by item. We surely believe that the quality of the paper will satisfy the requirements of this journal.

Round 2

Reviewer 1 Report

Authors have addressed the remarks made by the reviewer at a rather low extent. Although the language has been improved (in terms of grammar, syntax corrections), the terminology used and the selection of words are in many cases not appropriate. This comment is mostly referred to the electro-Fenton process and the respective mechanisms employed in aqueous solutions (for reference read the works published by the group of Brillas, e.g. Chemical Reviews 2009, 109, 6570–6631). Moreover, the reviewer has a serious concern about the inability of the authors to measure the concentration of H2O2. H2O2 could be measured easily, in the absence of ferrous ions (blank experiments of H2O2 electrosynthesis). It is logical not to measure H2O2 in the presence of iron catalyst, due to the induced Fenton reactions. The whole paper is based on H2O2 electrosynthesis and its effect on process performance, but no such data are presented herein. Other important remarks on the revised manuscript are as follows:

- The title is not  representative of the scope of the paper. Here, CNT were also used for modifying a 3-D carbon substrate (carbon felt).

- Line 46: Correct: "...electro-Fenton process for the treatment of wastewater of low to moderate organic strength". Ref. missing.

-Line 52: Correct "...degradation of dyes".

-Lines 53-54: irrelevant citation - why mentioning here the biological EF concept?

-Lines 56-57: Correct: "...The respective chemical reactions (Eq. 1-3) are are given below".

-Line 64: LDH?

-Line 71: Correct "...promotes the adsorption...".

-Lines 108-109: The novelty is still not clear. Authors have already commented on such improvements as described in previous works. 

-Line 115: Correct title "Experimental procedure".

-Line 117: Correct "...three-electrode".

-Lines 119-120: Correct "...the working electrode and the counter electrode...".

-Line 123: Provide a short justification of this optimum value. It is not understood why authors are based this selection on previous works. As mentioned below, the LSV measurement is not detrimental of such selection (no mass trasnfer current observed-meaning that no plateau of V vs I is measured here).

-Line 128: Correct "Schematic illustration...".

-Line 130: Correct title, e.g. Carbon felt electrode modification.

-Lines 131-134: Remove.

- Remove Tables 1, 2, 3, 4. All data are described in the manuscript (repetion of rather minimum results).

Author Response

Cover letter to editor

JournalMaterials (ISSN 1996-1944)

Manuscript IDmaterials-488606

Manuscript title: Application of Graphene Carbon Felt Electrodes in the Electro-Fenton System

TypeArticle

Submitted to SectionAdvanced Composites

Dear Editor

We appreciate the reviewers’ valuable comments and the editor’s help. The 2nd round revision has been done based on the 1st revision and according to all the reviewers’ comments. To clearly show the revisions, the revised manuscript keeping the “Track Changes” functions of MS word is also attached at the end of this cover letter. The manuscript has been carefully checked and revised again.

The original paper title “Application of Graphene Carbon Felt Electrodes in the Electro-Fenton System” has been slightly revised as “Application of Graphene and Carbon Nanotubes on Carbon Felt Electrodes for the Electro-Fenton System”.

The manuscript has been edited in the language by the professional English editing company “editage Co.”. An English editing certificate for editage Co. is attached to the end of this cover letter.

We surely believe that the quality of the paper will satisfy the requirements of this “Materials”. Your kindest consideration will be greatly appreciated.

I am looking forward to hearing from you.

With best wishes,

Yours sincerely

Yi-Ta Wang

Yi-Ta Wang

Division Director, Center for Nano Science & Technology.

Associate Professor,

Department of Mechanical and Electromechanical Engineering, National Ilan University NIU.

No. 1, Sec. 1, Shennong Rd., Yilan City, Yilan County 26047, Taiwan, R.O.C.

Tel: +886-3-9317457; +886-3-9317082

Reply to the Reviewer's comments:

Reviewer 1

Comment 1: Authors have addressed the remarks made by the reviewer at a rather low extent. Although the language has been improved (in terms of grammar, syntax corrections), the terminology used and the selection of words are in many cases not appropriate. This comment is mostly referred to the electro-Fenton process and the respective mechanisms employed in aqueous solutions (for reference read the works published by the group of Brillas, e.g. Chemical Reviews 2009, 109, 6570–6631). Other important remarks on the revised manuscript are as follows:

Reply 1We appreciate the reviewer’s comments. The manuscript has been carefully checked and revised again. The manuscript has been edited in the language and style by the professional English editing company “editage Co.”. An English editing certificate for editage Co. is attached to the end of this letter. And all the contents in the manuscript has been revised seriously according to the comments item by item.

The text “hydroxyl free radicals” have been revised to “hydroxyl radical” in the revision manuscript (line 39, 41, 44, 53, 252, 279, 305).

The text “•OH” have been revised to “OH” in the revision manuscript (line39, 45).

Comment 2: Moreover, the reviewer has a serious concern about the inability of the authors to measure the concentration of H2O2. H2O2 could be measured easily, in the absence of ferrous ions (blank experiments of H2O2 electrosynthesis). It is logical not to measure H2O2 in the presence of iron catalyst, due to the induced Fenton reactions. The whole paper is based on H2O2 electrosynthesis and its effect on process performance, but no such data are presented herein. Other important remarks on the revised manuscript are as follows:

Reply 2 Thanks for the reviewer’s useful and important comment. To satisfy the requirement for this section, we have studied some researches to understand how to measure the H2O2 during electrogeneration process; we referred the study (published by the group of 18.19.     Babaei e.g. Journal of Industrial and Engineering Chemistry 2017, 52, 270-276) and made experiment successfully. The content for introduction section has been revised, as below:

“When measuring the H2O2 yields for each electrode, the solution only contained 0.1 M KNO3 in the tank; titanium (IV) sulfate (Ti(SO4)2) was used as the reagent that reacted with the H2O2 produced during the electrogeneration process. The absorbance for the sample mixed with Ti(SO4)2 and H2O2 was measured using a visible-light spectrophotometer (wavelength: 410 nm), and the calibration curve was used to determine the concentration of H2O2 [19].” It has been revised in the revision manuscript (line 99-104).

“To confirm that the results of the LSV experiment were accurate, the H2O2 concentration was measured using the colorimetry method. Figure 4(B) shows the H2O2 concentration for different modifications of carbon felt via electrogeneration after 30 min; the graphene/C electrode generated the highest H2O2 concentration (0.261 mM), which was similar to that generated by the graphene foam electrode with 3D structures prepared by Mousset et al. [17]; the unmodified carbon felt and CNT/C electrode produced about 0.098 and 0.138 mM of H2O2 after 30 min, respectively.”. It has been revised in the revision manuscript (Line 193-198).

The graph “H2O2 concentration for different modifications of carbon felt via electrogeneration after 30 min (0.65 V) has been revised in the revision manuscript (Figure 4(B) line 202-203).

[19] Babaei-Sati, R.; Basiri Parsa, J. Electrogeneration of H2O2 using graphite cathode modified with electrochemically synthesized polypyrrole/MWCNT nanocomposite for electro-Fenton process. J. Ind. Eng. Chem. 2017, 52, 270-276, 10.1016/j.jiec.2017.03.056.

Comment 3: The title is not representative of the scope of the paper. Here, CNT were also used for modifying a 3-D carbon substrate (carbon felt).

Reply 3Thanks for the reviewer’s comment. The tittle has been revised “Application of Graphene and Carbon Nanotubes on Carbon Felt Electrodes for the Electro-Fenton System” in the revision manuscript.

Comment 4: Line 46: Correct: "...electro-Fenton process for the treatment of wastewater of low to moderate organic strength". Ref. missing.

Reply 4Thanks for the reviewer’s comment. It has been revised in the revision manuscript (line 33-35).

Comment 5: Line 52: Correct "...degradation of dyes".

Reply 5 Thanks for the reviewer’s comment. It has been revised in the revision manuscript (line 40).

Comment 6: Lines 53-54: irrelevant citation - why mentioning here the biological EF concept?

Reply 6 Thanks for the reviewer’s comment. It has been revised in the revision manuscript (line 41-42).

Comment 7: Lines 56-57: Correct: "...The respective chemical reactions (Eq. 1-3) are given below 

Reply 7 Thanks for the reviewer’s comment. It has been revised in the revision manuscript (line 44-45).

Comment 8: Line 64: LDH?

Reply 8 Thanks for the reviewer’s comment. It has been revised “CoFe-layered double hydroxide (CoFe-LDH)” in the revision manuscript (line 51).

Comment 9: Correct "...promotes the adsorption..."

Reply 9 Thanks for the reviewer’s comment. It has been revised “and the surface of CNT promotes the adsorption of hydrocarbons” in the revision manuscript (line 58).

Comment 10: Lines 108-109: The novelty is still not clear. Authors have already commented on such improvements as described in previous works.

Reply 10Thanks for the reviewer’s important comment. It has been revised “According to the literature cited above, CNTs and graphene have been widely used as electrode materials; however, hardly any research has demonstrated the difference between the efficiency of CNTs and graphene material in the electro-Fenton system. In this study, carbon felt was used as the substrate, and CNTs and graphene were used to modify the electrodes; this was expected to increase the specific surface area and the property for oxidation-reduction reactions of the cathode in the electro-Fenton system. The effects of the modified carbon felt material on the cathode plates were investigated.” in the revision manuscript (line 83-89).

Comment 11: Line 115: Correct title "Experimental procedure".

Reply 11 Thanks for the reviewer’s comment. It has been revised in the revision manuscript (line 91).

Comment 12: Line 117: Correct "...three-electrode".

Reply 12 Thanks for the reviewer’s comment. It has been revised in the revision manuscript (line 93).

Comment 13: Lines 119-120: Correct "...the working electrode and the counter electrode..."

Reply 13 Thanks for the reviewer’s comment. It has been revised in the revision manuscript (line 95).

Comment 14: Line 123: Provide a short justification of this optimum value. It is not understood why authors are based this selection on previous works. As mentioned below, the LSV measurement is not detrimental of such selection (no mass trasnfer current observed-meaning that no plateau of V vs I is measured here).

Reply 14Thanks for the reviewer’s comment. The range for carbonaceous is about -0.6V~ -0.7 V, Which has been reported in some studies; the system will cause “H2 evolution” when the potential at -0.65 V, and it influences the production for H2O2 [17, 18]. In LSV section, all the electrodes showed that change of response current rose gradually at potential=-0.65 V, indicated the “H2 evolution” started to react.

Ref  

[17] Mousset, E.; Wang, Z.; Hammaker, J.; Lefebvre, O. Physico-chemical properties of pristine graphene and its performance as electrode material for Electro-Fenton treatment of wastewater. Electrochim. Acta. 2016, 214, 217-230.[18] Wang, Y.; Liu, Y.; Wang, K.; Song, S.; Tsiakaras, P.; Liu, H. Preparation and characterization of a novel KOH activated graphite felt cathode for the Electro-Fenton process. Appl. Catal. B 2015, 165, 360-368

Comment 15: Line 128: Correct "Schematic illustration..."

Reply 15 Thanks for the reviewer’s comment. It has been revised in the revision manuscript (line 108).

Comment 16: Line 130: Correct title, e.g. Carbon felt electrode modification.

Reply 16 Thanks for the reviewer’s comment. It has been revised in the revision manuscript (line 110).

Comment 17: Lines 131-134: Remove.

Reply 17 Thanks for the reviewer’s comment. It has been revised in the revision manuscript (line 111-114).

Comment 18: Remove Tables 1, 2, 3, 4. All data are described in the manuscript (repletion of rather minimum results).

Reply 18 Thanks for the reviewer’s comment. It has been revised in the revision manuscript (line 109, 124-125, 172, 223).

Reviewer 3 Report

All the issues have been addressed properly. I recommend the publication of this manuscript.

Author Response

Cover letter to editor

JournalMaterials (ISSN 1996-1944)

Manuscript IDmaterials-488606

Manuscript title: Application of Graphene Carbon Felt Electrodes in the Electro-Fenton System

TypeArticle

Submitted to SectionAdvanced Composites

Dear Editor

We appreciate the reviewers’ valuable comments and the editor’s help. The 2nd round revision has been done based on the 1st revision and according to all the reviewers’ comments. To clearly show the revisions, the revised manuscript keeping the “Track Changes” functions of MS word is also attached at the end of this cover letter. The manuscript has been carefully checked and revised again.

The original paper title “Application of Graphene Carbon Felt Electrodes in the Electro-Fenton System” has been slightly revised as “Application of Graphene and Carbon Nanotubes on Carbon Felt Electrodes for the Electro-Fenton System”.

The manuscript has been edited in the language by the professional English editing company “editage Co.”. An English editing certificate for editage Co. is attached to the end of this cover letter.

We surely believe that the quality of the paper will satisfy the requirements of this “Materials”. Your kindest consideration will be greatly appreciated.

I am looking forward to hearing from you.

With best wishes,

Yours sincerely

Yi-Ta Wang

Yi-Ta Wang

Division Director, Center for Nano Science & Technology.

Associate Professor,

Department of Mechanical and Electromechanical Engineering, National Ilan University NIU.

No. 1, Sec. 1, Shennong Rd., Yilan City, Yilan County 26047, Taiwan, R.O.C.

Tel: +886-3-9317457; +886-3-9317082

Reviewer 3

Comment 1: English language and style are fine/minor spell check required

Reply 1Thanks for the reviewer’s comment. The manuscript has been carefully checked and revised again. The manuscript has been edited in the language and style by the professional English editing company “editage Co.”. An English editing certificate for editage Co. is attached to the end of this letter.

Comment 2All the issues have been addressed properly. I recommend the publication of this manuscript.

Reply 2Thanks for the reviewer’s positive comment.
